# The Burden of Treatment: Experiences of Patients Who Have Undergone Radiotherapy and Proton Beam Therapy

**DOI:** 10.3390/healthcare13111351

**Published:** 2025-06-05

**Authors:** Danielle Fairweather, Rachel M. Taylor, Lee Hulbert-Williams, Nick J. Hulbert-Williams, Rita Simões

**Affiliations:** 1Cancer Division, University College London Hospitals NHS Foundation Trust, London NW1 2BU, UK; rita.simoes@nhs.net; 2Department of Targeted Interventions, University College London, London W1W 7TS, UK; 3Department of Psychology, Edge Hill University, Liverpool L39 4QP, UKnick.hulbert-williams@edgehill.ac.uk (N.J.H.-W.)

**Keywords:** patient experience, radiotherapy, proton beam therapy, toxicity, quality of life, qualitative research

## Abstract

**Background/Objectives**: The physical and psychosocial impacts of cancer treatment can be distressing and profound for many patients, but little is known about the specific impacts of undergoing radiotherapy and proton beam therapy (PBT). This study explores the hidden burdens of treatment and aims to identify the gaps in our current understanding of patients’ experience when attending a large radiotherapy and PBT service. **Methods**: A qualitative study using semi-structured interviews was conducted with patients undergoing treatment. A purposive sample of participants were recruited, reflecting the main indications for radiotherapy and PBT. Semi-structured interviews were conducted between August 2023 and January 2024 either in person, virtually, or by telephone. Data were analysed using Framework Analysis. **Results**: In total, 20 patients were interviewed. Five themes were identified: informational needs, emotional wellbeing, logistical concerns, physical impacts, and interpersonal impacts. Patients reported additional financial burdens such as transport and staying away from home, difficulty carrying out normal responsibilities, caregiver burden, and increased anxiety. Many patients reported the post-treatment drop in healthcare interaction, which resulted in distress and isolation, difficult. **Conclusions**: This study indicates that there are many burdens of radiotherapy and PBT outside of the physical symptoms and side-effects of cancer treatment. Tailored support is needed to address treatment-specific concerns within the radiotherapy and PBT service, but this study also suggests that supportive interventions developed for broader cancer populations may be helpful for this patient cohort.

## 1. Introduction

It is estimated that 40% of all cancer patients who survive 5 years receive radiotherapy as part of their treatment plan [1]. Many patients receiving radiotherapy or proton beam therapy (PBT) experience substantial physical and psychosocial challenges, which can be deeply distressing [2]. Beyond the physical side-effects of treatment, patients and caregivers often deal with hidden burdens such as logistical, financial, and interpersonal impacts, which are frequently overlooked in favour of survival [2,3]. Logistical concerns such as managing time off work or school, financial pressures such as the direct costs of transport or reduced income, and interpersonal impacts such as shifts in family dynamics or roles within the family have all previously been reported but rarely taken into account when exploring cancer treatment toxicity [2,3].

While photon radiotherapy is the most common form of radiotherapy, PBT is an alternative which uses protons instead of photons [4]. This offers improved dosimetry, allowing for more precise targeting of tumours, which can result in reduced long-term side-effects [4,5]. Although the appearance of gantry rooms in PBT facilities is quite different from that of those used in conventional photon radiotherapy, the true value of PBT lies in the radiobiological advantages of protons [6]. These include their ability to deliver highly targeted doses of radiation while reducing doses to surrounding healthy tissues [5].

This has particular value for children, reducing long-term late effects, and in adults with radio-resistant tumours close to critical structures [6,7]. Despite these advantages, current United Kingdom (UK) provision allows only 1% of the population requiring radiotherapy to access PBT [6]. There are currently only two public healthcare centres—London and Manchester—that provide PBT in the UK [6,8]. Therefore, those having to travel from outside of these cities for PBT will potentially face additional costs and the strain of staying away from home [9].

Although the cost of accommodation and the PBT treatment is covered by NHS England for patients who meet the indication criteria for PBT, patients may have to cover other costs such as travel and food for up to eight weeks of treatment in an unfamiliar metropolitan city [6]. No data are currently available about these out-of-pocket costs [10]. Travelling while sick can also be challenging, and patients may also have to leave family and support systems at home for the duration of their treatment. This combination may reduce patient compliance or make patients opt for traditional photon radiotherapy which is more readily available in closer local hospitals [9,10].

There has been considerable research within radiation oncology over recent years that aims to reduce physical treatment-related toxicities [11,12,13,14,15,16]. Improvements in treatment delivery and imaging, adjusting treatment plans in real time, and symptom management have all been individually explored and, in some cases, add substantial value in the management of certain cancers [2,5,6,17]. However, there has been less research investigating the lived experience of patients undergoing radiotherapy and PBT, including the actual treatment delivery process itself [18,19,20]. It is also important to note that even when treatment-related toxicity is reduced, there is no guarantee this in itself will also lead to improvements in patient experience.

Hypofractionation is an example of this. Traditionally, radiotherapy is delivered across a number of ‘fractions’, requiring daily hospital visits to allow the radiation dose to accumulate. Radiotherapy for low-risk prostate cancer has over time been reduced from ~39 fractions (8 weeks) down to 20 fractions (4 weeks). Following on from recent research there has now been another shift towards five fractions (5 days over 1–2 weeks) [21]. Though radiotherapy clinical trials do compare patient outcomes between these treatments, little research explores the patient perspective on reduced fractionation [22]. At the other end of the spectrum, radio-resistant tumours like sarcomas still require high doses of radiotherapy or PBT, necessitating prolonged treatment durations of 6–8 weeks [6]. Existing qualitative studies which do explore patient perspective often focus on information needs, survivorship, or symptom assessment [3,18], with limited attention to the holistic patient experience and perceived burden [2,3].

Failing to address hidden toxicities can directly, or indirectly, result in delayed or aborted treatments, with suboptimal results, which furthers the propagation of inequalities [2,3]. Recent research from Switzerland found that some patients refused to have PBT, even when beneficial to their long-term outcomes, due to the associated burden of staying away from home [10]. Patient experience is a fundamental measure of the quality of care, and reviewing patient feedback is recommended for identifying and addressing the unseen impacts of treatment [23,24]. While there is a national patient experience survey that is used in radiotherapy centres in the UK, qualitative research can be used to further investigate and understand the causes of experiences and perceptions in patients [24,25].

The photon radiotherapy and PBT service at our large inner-city university hospital operates with a shared, rotational workforce and offers comparable levels of clinical support to patients across both treatment modalities [6]. Importantly, there is often clinical overlap between the two pathways; for example, patients receiving PBT frequently have a backup photon radiotherapy plan in place in case of machine breakdowns or other technical issues [6]. Given this interconnectedness, it was important to explore both treatment experiences to gain a comprehensive understanding of patient journeys within the service. Therefore, this study aimed to explore and understand the experiences of patients receiving radiotherapy and PBT at a large inner-city university hospital service. From this information, we identify the full spectrum of needs across the services and make recommendations on where to prioritise service improvement.

## 2. Materials and Methods

This was a qualitative study that used semi-structured interviews conducted with patient participants. Ethical approval was granted by the Health Research Authority (23/PR/0549). Interviews were conducted between August 2023 and January 2024.

### 2.1. Participants

Purposive sampling was used to reach maximum variation across the included diagnoses to ensure heterogeneity of views across the different experiences of treatment. The aim was to recruit 30 patients, as a sample size above 20 is recommended to achieve data saturation in qualitative research [26]. This target also allowed for a buffer to account for potential participant drop-out or those who later declined to be interviewed post-treatment.

Participants were included if they were ≥16 years old; diagnosed with brain, breast, head and neck, lung, or prostate cancer, or sarcoma; and, able to speak and understand English. All patients were required to have localised disease in order to participate, as metastatic stage may have a stronger influence on wellbeing regardless of treatment type [27].

Participants were identified as eligible by their healthcare teams while on treatment. After receiving the patient information sheet, potential participants met with the researcher (DF) who explained the study and answered any questions. Written consent was obtained prior to participation. Patients were interviewed 2–6 weeks after finishing radiotherapy or PBT, allowing for the peak side-effects (7–10 days) but also minimising recall bias.

### 2.2. Procedure

A semi-structured interview schedule was developed from the existing literature [28,29] and consisted of questions that explored care experience, physical health, emotional health, social wellbeing, and symptom management. The interview schedule aimed to cover the common facets of quality of life [29] as well as explore the specific experience of treatment delivery. The questions were tested with a volunteer participant with lived experience. Patients were offered interviews by telephone, video call via Microsoft Teams, or in person at the hospital. They were recruited and consented during their treatment, then contacted two weeks post-treatment to reconfirm their interest and schedule the interview. All interviews, conducted by the same researcher (DF), were digitally recorded and transcribed using AI software (Otter.AI: version 3.5.0).

Otter AI was chosen due to its high reported transcription accuracy for English-language recordings in healthcare and research settings [30]. However, recognising that automated transcription tools can be limited in distinguishing between speakers, handling overlapping dialogue, and capturing medical or technical terminology, each transcript was manually reviewed and edited to ensure verbatim accuracy [30,31]. Transcripts were anonymised and verified by one researcher and independently reviewed by two additional team members to confirm reliability and minimise transcription errors. Audio recordings were deleted following validation of each transcript.

### 2.3. Data Analysis

Data were analysed using Framework Analysis, a five-stage process starting with familiarisation [25]. The theoretical framework was developed inductively from the WHO-QoL-100 [29] and National Health Service (NHS) patient experience framework [32] in Microsoft Excel (Table 1). Additional themes were added deductively from the transcripts. One researcher coded the transcripts against domains and facets, and this was reviewed for inter-coder reliability by two other members of the research team. Discrepancies were discussed and resolved collaboratively to ensure consistency in interpretation and coding.

Constant comparative analysis was used to apply deductive codes across the data. Transcripts were charted in Microsoft Excel according to the indices in the framework. The research team then reviewed and discussed the findings to map and interpret the main themes. Investigator triangulation was employed to increase the credibility of the research findings. Data saturation was actively monitored throughout the analysis process and was considered achieved when no new themes or insights emerged from the final interviews, supporting the adequacy of the sample size.

## 3. Results

In total, 30 patients receiving radiotherapy or PBT with curative intent were approached for the study; 10 subsequently declined, and 20 patients completed the interviews. The diagnosis and treatment site of the participants is reported in Table 2. The patients were aged 25–76 years, mainly Caucasian (n = 16, 80%), and lived between 2.4 and 168.0 miles (average = 24.72 miles) from the hospital.

Five themes were identified in the analysis (Figure 1): informational needs, emotional wellbeing, physical impacts, logistical concerns, and interpersonal impacts. Supporting quotes illustrate key themes, with additional quotes presented in the Appendix A.

### 3.1. Informational Needs

The first theme that was explored was the informational needs of patients. The information given before treatment, such as side-effects and education on the treatment process and preparation, was mostly deemed acceptable to both the PBT and radiotherapy patients. A few patients continued to seek further information through the likes of charity-run initiatives; however, a minority felt overloaded by the information:


*‘He (the consultant) went through all the side effects and when I came out of that, I remember thinking Oh my God I just don’t know if I wanted to hear all of that’, but they have to say it all of course. Sometimes you hear so much that it worries you more than anything else.’ Breast sarcoma, radiotherapy.*


The lack of support and guidance that was available for patients after they finished treatment was highlighted as an area that required improvement irrespective of cancer type. Some patients reported a lack of understanding about what happens post-treatment, as well as the need for further guidance on returning to everyday activities:


*‘You end up doing it on your own, without any guidance.’*
Prostate cancer, radiotherapy.

### 3.2. Emotional Wellbeing

This theme encompassed the psychological and emotional wellbeing of patients. All patients reported an increase in emotional symptoms such as anxiety and distress when receiving treatment irrespective of cancer diagnosis. There were periods where these feelings were more heightened than others, including before or mid-treatment, when symptoms began to worsen, and at the end of treatment, when symptoms were worst. Uncertainty of the future was also a cause for increased distress and anxiety as patients wondered if the treatment had worked or what the future would hold for them:


*‘Because like I said, there were some dark moments. And I suppose the biggest one is I don’t know what lies ahead.’*
Sarcoma, radiotherapy.

In spite of this, the majority of patients also reported having hope and optimism during the treatment process and for the future. These patients tended to be aware that their cancer diagnosis had a good prognosis and was deemed to be curable. They also tended to focus on trusting professionals and feeling like they were in a fortunate position. Patients who received PBT remarked that they were grateful to have been able to receive what was perceived to be the best treatment option for their diagnosis:


*‘I appreciate how lucky and I have been right, so there’s no denying that in terms of the physical symptoms I had it easy, I got scanned quickly, got referred quickly to (city name), had an amazing world expert deal with it, lucky enough to have like superb proton beam therapy which very few people get all that say.’*
Brain tumour, PBT.

To counter declining emotional wellbeing, some patients drew on support from other areas of life to get through treatment, one of those areas being faith and spirituality. For those who identified as religious or spiritual, leaning on their belief system was an important part of their experience, and sometimes they required additional support from healthcare professionals:


*‘I asked them for a longer gown. Because I didn’t like you know, I mean in my religion, if you just ask them, they will do it. They will make sure you are comfortable.’*
Sarcoma, radiotherapy.

Another common subtheme was the impact of treatment on body image, which resulted from a change in appearance, e.g., hair loss, both facial hair and head hair, and skin changes. Changes in weight (loss or gain) were emotionally distressing and affected self-esteem and sexual functioning:


*‘I don’t feel like as sexual as before, so I’m hoping that comes back. I’m hoping sort of like, you know, builds up, and maybe you know, that will happen once I start exercising (losing weight) and feeling better about myself.’*
Breast cancer, radiotherapy.

### 3.3. Physical Impacts

There were many physical impacts of treatment experienced. There were those related to treatment-site-specific side-effects, e.g., difficulty swallowing and oral pain with the patients receiving head and neck radiotherapy. The most common symptoms highlighted across all participants were fatigue or tiredness and skin soreness. Most physical symptoms were not noticed initially but worsened as they got closer to finishing treatment before peaking in severity post-treatment:


*‘From two to three, I start to feel I’m getting sore and by then by the time I got to week three, I stopped eating. Week four my mouth was just horrendous I couldn’t eat anything because just I just wanted to be sick I couldn’t swallow.’*
Head and neck cancer, radiotherapy.


*‘It actually hit me about roughly about a week afterwards. And then there was a period of about one week where I was in bed most of the time.’*
Head and neck cancer, PBT.

Patients receiving prostate radiotherapy emphasised the difficulty of managing urinary and bowel symptoms around treatment preparation, specifically around transport and the need to be close to a toilet when travelling to and from the hospital:


*‘Like five weeks into the treatment I was. I didn’t get caught short. I was very close to it at the train station sometimes because it in a heartbeat is like you need to go to the toilet. And it’s like, Where the hell’s the toilet?’*
Prostate cancer, radiotherapy.

Physical comfort during treatment was noted by patients who were required to use immobilisation masks. There was discomfort associated with the mask, specifically in the MRI scan required for PBT:


*‘Oh, it’s horrible… It was the fact that I had to like they said I had to lie perfectly still without moving right. And that’s fine for a few minutes. But it actually took 40 minutes. Right. I had to lie completely still not move at all for 40 minutes with my head clamped down. And after at the end of it, my whole body had sort of gone numb.’*
Head and neck cancer, PBT.

Patients receiving treatment for breast or lung cancer reported having to hold an uncomfortable position with their arms above their head. The exposure of the chest was also associated with discomfort and vulnerability, as well as the need for tattoos:


*‘For me, lying down on the table with my arms above my head was, was an uncomfortable feeling. It was, it’s a sense of vulnerability… It’s difficult to get comfortable. You have to hold it for a while. Worst thing.’*
Breast cancer, radiotherapy.


*‘I didn’t like having tattoos. And I was very upset about that… And I don’t think it’s very nice to give people tattoos. I think that’s emotionally quite disturbing. I think it’s cruel.’*
Breast cancer, radiotherapy.

Reduced sexual functioning was experienced by patients receiving treatment to the prostate and breast, including reduced libido. In male patients, reduced erection capacity was mentioned as the common treatment effect, but none of the patients had explored management options with a healthcare professional:


*‘I can’t gain an erection at the moment. But, you know, if it does become a big problem, then then perhaps I’ll ask for some help. But at the moment, I’ve got bigger fish to fry at the moment.’*
Prostate cancer, radiotherapy.

The physical impacts of treatment also had effects on the daily lives and activities of patients. Patients who experienced tiredness and fatigue, as well as other physical symptoms, found they had a reduced capacity to perform daily activities.


*‘And I’ve not got back to walking. I used to spend a lot of time walking before.’*
Head and neck cancer, radiotherapy.

### 3.4. Logistical Concerns

The reduced capacity for daily activities was also reported as a logistical concern and exacerbated by either having to stay away from home or spend large amounts of the day travelling to the hospital:


*‘You can’t really plan anything on any given day, because it’s for me, obviously, it was travelling into London every day at different times every day. Yeah, absolutely. It was almost having to write off five weeks, but I couldn’t really think of a way around that.’*
Sarcoma, radiotherapy.

As well as the logistics of fitting life around treatment, there were concerns surrounding transport and accommodation. During the period when participants were on treatment, there was a notable amount of industrial action in the UK on public transport. Transport was therefore highlighted as a common logistical concern amongst patients, which also had a large financial burden too:


*‘The biggest issue that I faced was the railways. My son in law drove up to (city name) to pick me up because it was no way I was gonna get home for at least another five hours.’*
Prostate cancer, radiotherapy.


*‘The first week of radiotherapy I did use mini cabs. I spent about two to 300 quid so I couldn’t go on like this. So yeah, I went on hospital transport.’*
Lung cancer, radiotherapy.

Patients who were travelling long distances were given accommodation for the duration of their treatment, but they also highlighted concerns about being away from home, specifically the financial burden of this:


*‘But obviously, you have to stay up there (city name) and we came over the weekend. And obviously, the travel costs. Yeah, we had to fork out which is a lot of money. The policy decision is made across the UK is not to help the patient’s travel to Manchester or London. Unless they did receive a qualifying benefit, which is very difficult these days. Yeah, that’s, you know, you shouldn’t need to worry about that, the financial drain of travelling back and forth.’*
Sarcoma, PBT.

Another logistical concern was the coordination and accessibility of care. The centre is a specialist referral centre for PBT and sarcoma, so care for these patients was often delivered across multiple organisations. At times this resulted in patients not knowing whom to contact or having to personally coordinate healthcare teams:


*‘The NHS doesn’t seem to have one NHS databank. For example, I was having to forward emails from one hospital to another hospital because they weren’t kept in the loop, or they weren’t aware of each other if you’d like.’*
Sarcoma, radiotherapy.

Some patients opted to use private medical insurance to access PBT after being unable to receive it through the NHS, but this also came with the need for additional coordination:


*‘I felt as if I was coordinating amongst all the professionals because I required in order neurology, endocrinology and kind of radiologists for the proton beam therapy, so that in itself was stressful.’*
Brain tumour, PBT.

### 3.5. Interpersonal Impacts

The last theme reflected the impact that treatment had on the patients’ relationships. It was recognised that there was both a physical and psychosocial caregiver burden experienced by family members who were supporting them through treatment. There were five men with differing diagnoses who highlighted that they tried to protect their partners from the emotional toll of their cancer:


*‘You know, I’m not hiding my sort of true results or anything. It’s just I don’t want to worry her. I filter things.’*
Sarcoma, radiotherapy.

A further interpersonal impact was patients establishing peer support with other patients. Speaking to someone who was going through a similar experience brought comfort as patients moved through this period of uncertainty:


*‘I met other people that were going through the same thing with me. But obviously they had different areas. I met this lady who had the same time as me on the same place. Just has a little bit up on her leg. It was just nice to meet them, you know just to talk to them.’*
Sarcoma, radiotherapy.

Lastly, patients built strong rapport and relationships with their healthcare teams as they moved through the treatment process. Radiographers were noted to be supportive, comforting, and reassuring. The consistency of being treated by the same team daily and building a strong bond with radiographers helped patients get through their treatment:


*‘They play a very important part to make the whole experience feel, yeah, so friendly. So actually, after like, one week, one week’s time, you already don’t feel nervous. Yeah, and very happy to see the similar face.’*
Breast sarcoma, radiotherapy.

## 4. Discussion

This is the first study exploring the holistic experiences of patients undergoing radiotherapy and PBT at a large inner-city hospital. The participants reported concerns specific to radiotherapy or PBT, such as daily travel burden, distress from radiotherapy tattoos, and discomfort from immobilisation. However, financial strain, caregiver burden, and emotional distress align with experiences reported in broader cancer research [3]. Neglecting hidden burdens can delay or disrupt treatment, worsening outcomes and healthcare inequalities [2,3]. While some challenges are unique to radiotherapy or PBT and require targeted solutions, existing interventions for common cancer-related impacts may also benefit these patients.

A common identified subtheme was the impact on appearance. Radiotherapy can negatively impact body image, which can result in social appearance anxiety [33]. The permanent tattoo marks, given to position patients correctly for treatment delivery, serve as a constant reminder of cancer diagnosis/treatment as well as affecting clothing choices and body confidence [34]. Avoiding unpleasant visual stimuli is a common coping strategy that helps emotional regulation [35]. The visual stimuli of radiotherapy tattoos could dysregulate this coping strategy, which could explain the distress experienced by these patients. A positive association between worse body image scores and skin toxicity has also been previously reported [33].

Weight gain can also have a negative impact on body image in patients with cancer [36]. Patients receiving radiotherapy and PBT treatment for breast and prostate cancer in this study reported weight gain as a concern, but it was not clear whether this related specifically to treatment given that patients also attributed it to hormone therapy and inactivity due to treatment fatigue. Physical activity interventions and cognitive–behavioural therapy have been suggested by previous researchers to improve body image in the broader population of cancer patients [37]. Our findings suggest that this approach may also be applicable to patients receiving radiotherapy and PBT.

Our study highlighted that the treatment delivery process can also be distressing. Patients have previously reported that positioning and discomfort during radiotherapy can impact emotional health and result in increased anxiety and fear [20]. Female patients in this study felt especially vulnerable when exposing their chest, which has previously been described in a qualitative study exploring the experiences of patients receiving radiotherapy for breast cancer [18].

Patients receiving treatment for head and neck cancer also found the immobilisation masks distressing, particularly during MRI scans for PBT treatment planning. The concept of ‘mask anxiety’ has been previously defined in qualitative studies exploring patient experience in head and neck cancer, but these studies specifically looked at photon radiotherapy, and this issue has yet to be explored in PBT, where MRI scans are routinely utilised [38,39]. Many comfort interventions have been explored to improve comfort and reduce distress. Examples such as listening to music, coaching, and aromatherapy have all been explored with limited amounts of favourable evidence [40]. A combined ‘comfort package’ may be most effective in reducing anxiety, but its effectiveness remains unreported [40].

Anxiety was also often coupled with fears about the future and uncertainty over symptom progression. Illness anxiety has previously been studied in cancer patients, with informational and emotional support being proven to have a positive impact on quality of life [41]. Many cancer patients feel ‘abandoned’ after treatment ends [42], a concern echoed by our participants who also sought more post-treatment support. Therapeutic radiographers, with their unique technical experience and compassionate care skills, are well equipped to support these patients [43]. Continued contact could provide patients with the additional guidance that they need, though it is not currently standard of practice [42].

Our patients perceived that their caregivers also face additional burdens, not just emotionally, but physically with tasks like housework, childcare, and financial support. While caregivers often share the emotional strain of a cancer diagnosis, they typically receive less support than patients [3,44]. Notably, we found that male patients tended to shield loved ones from their true feelings, which may in part be a way of protecting them. Previous research suggests that men are less likely to accept emotional support from healthcare professionals [45]. This study suggests this carries through into their personal lives, as they also do not want to burden caregivers with their emotions.

Concerns about the financial burden of treatment was universal across all treatment sites. Having PBT or being referred to a regional specialist team also came with the additional challenge of being away from home. The financial cost of cancer is well documented and can impact on patient outcomes [3,9]. The financial burden associated with PBT has been reported previously in Switzerland, with patients needing to cut back on expenses in order to save money or use savings to account for the cost of travelling [10]. Financial burden has also been documented as a reason patients refuse PBT and choose photon radiotherapy which is readily available closer to home [9,10]. This study expands on this prior knowledge by demonstrating that similar financial burdens are also seen within the UK.

### Study Limitations

This study had a number of limitations. The interviewer was part of the radiotherapy team, and although she did not directly interact with the participants outside of their research role, this may have introduced reporting bias. There is potential for selection bias, as only English-speaking participants were included, and although there was heterogeneity of religion and ethnicity in the participant sample, we may not represent the experience of patients who are non-English speakers. Participants were recruited from a single centre, so some findings could be unique to this service and therefore limit the ability to generalise findings to other populations and services. Despite approaching under 25s to participate in this study, none proceeded to interview. Therefore, the perspectives of adolescents and young adults are not represented. Future research should focus specifically on young people, as it is well recognised that they have healthcare needs different to children and older adults [46].

## 5. Conclusions

Despite limitations, this is the first in-depth report of patients’ experiences of radiotherapy and PBT. Identifying hidden treatment burdens can help improve adherence and patient-reported outcomes [3]. Radiotherapy and PBT involve daily contact, followed by a sharp decline in support. The financial and psychosocial support given to patients while undergoing treatment and post-treatment requires further improvement. While treatment-specific concerns must be addressed, our study also suggests that interventions developed for broader cancer populations may also be applied. Support needs vary, as some patients feel they have ‘bigger fish to fry’.

Still, these insights will directly inform the design of the service by guiding the development of more patient-centred care pathways, such as improved communication strategies, enhanced psychosocial support, and more flexible scheduling to reduce travel and waiting times. At a policy level, the findings support the case for greater investment in equitable access to advanced radiotherapy technologies like PBT and for policies that recognise the holistic burden of treatment, not just clinical outcomes.

## Figures and Tables

**Figure 1 healthcare-13-01351-f001:**
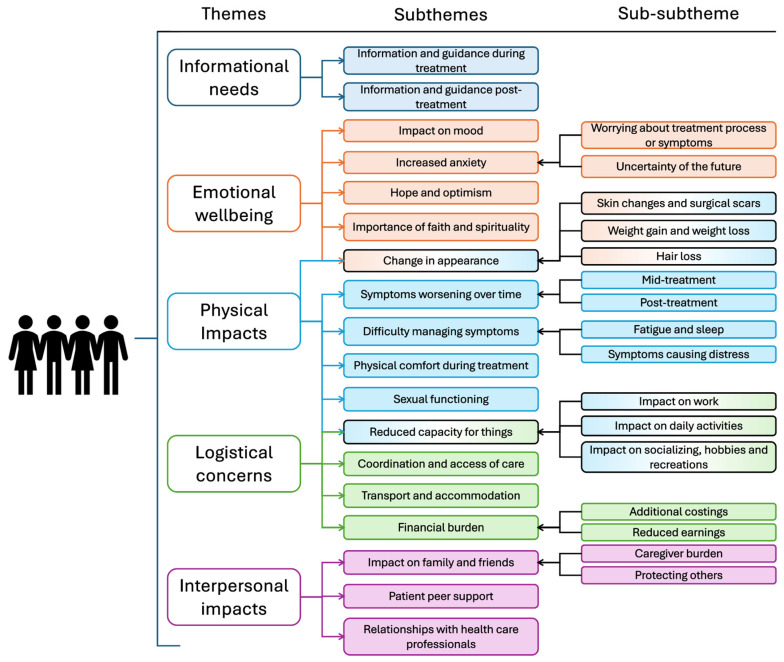
Themes and subthemes of patient experience of radiotherapy and proton beam therapy. Subthemes which are shaded with two different colours are found within two separate themes and are therefore highlighted as such (e.g., change in appearance affects emotional wellbeing as well as being a physical impact).

**Table 1 healthcare-13-01351-t001:** Data analysis framework.

Main Code	Sub-Code
Physical Symptoms	Pain and discomfort
	Vitality and fatigue
	Sexual activity
	Sleep
	Motor functioning
	Mobility
	Sensory functions
Psychological Function	Cognitive functioning
	Happiness and contentment
	Depression
	Anxiety
	Hopefulness and optimism
	Self-esteem
	Self-efficacy
	Body image
Levels of Independence	Ability to carry out activities of daily living
	Dependence on substances
	Communication capacity
	Working capacity
	Participation in and opportunity for recreation and pastimes
Social Relationships	Isolation/social contact
	Family support
	Support from friends/acquaintances
	Activities as provider/supporter
Environment	Freedom, physical safety, and security
	Quality of home environment
	Quality of work environment
	Work satisfaction
	Opportunities for acquiring new knowledge and skills
	Financial status
	Availability/access to health and social care
	Transport
	Religion/spirituality
Care Delivery	Coordination of care
	Information, communication, and education
	Physical comfort
	Emotional support
	Involvement of family and friends
	Transition and continuity
	Access to care

**Table 2 healthcare-13-01351-t002:** Participant characteristics.

Attribute	*n*	Mean (SD)
Age (All)	20	53.2 (14.0)
		Percentage (%)
Female	8	40
Male	12	60
Diagnosis		
Brain tumour	2	10
Breast cancer	2	10
Head and neck cancer	3	15
Lung cancer	2	10
Prostate cancer	4	20
Sarcoma	7	35
Treatment Site		
Brain	3	15
Breast	4	20
Head and neck	3	15
Lung	3	15
Male pelvis/prostate	4	20
Limb	3	15
Radiotherapy	15	75
Proton Beam Therapy	5	25
Treatment Prescription ^1^		
60Gy in 30 #	9	45
60Gy in 20 #	1	5
66Gy in 33 #	1	5
78Gy in 39 #	2	10
50Gy in 25 #	2	10
50.4Gy in 28 #	3	15
26Gy in 5 #	2	10

^1^ Radiation dose (Gy) delivered in daily fractions (#), 5 fractions delivered per week. Gy = Gray, SD = Standard Deviation, # = Fractionation.

## Data Availability

The original contributions presented in this study are included in the article/Appendix A. Further inquiries can be directed to the corresponding authors.

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
