# Peer review of "The Burden of Treatment: Experiences of Patients Who Have Undergone Radiotherapy and Proton Beam Therapy"

_healthcare, 2025, doi:10.3390/healthcare13111351_

Round 1

Reviewer 1 Report

Comments and Suggestions for Authors
  1. Title: The burden of treatment: experiences of patients who have undergone radiotherapy and proton beam therapy

    Manuscript ID: healthcare-3570444

    Review questions to author

    1. The study addresses an important and underexplored area the physical, emotional, and logistical burdens experienced by patients undergoing radiotherapy and proton beam therapy (PBT). The objective is clearly stated and relevant for clinical practice, health services, and psycho-oncology.

    2.Introduction
    While the abstract and conclusion indicate a strong rationale, the introduction in the main text should ensure that:

    • The gap in literature is clearly established.
    • Differences between PBT and conventional radiotherapy are highlighted.
    • The need for qualitative exploration is justified.
    1. Methodology Suggestions:
      • Clarify sample size justification.
      • Include ethical approval
      • Define inclusion and exclusion criteria.
      • Explain how themes were derived and validated.
    1. Results clearly presented and supported by data?
    • Five well-defined themes are presented.
    • Findings are rich and nuanced, e.g., issues of emotional wellbeing, financial stress, and abrupt decline in healthcare contact post-treatment.

    Suggestion: Include patient quotes in the results section of the full text to enhance the authenticity of the themes.

    1. Discussion and conclusions

    The conclusions are grounded in the data and emphasize practical implications like:

    • Tailored supportive care.
    • Potential for broader application of interventions.
    • Need to maintain continuity in patient interaction post-treatment.

    The tone is balanced, acknowledging the study’s novelty and limitations.

    1. Limitation

    Stated that this is the first in-depth report on the topic, suggesting novelty.

    However, the limitations section could elaborate on:

    • Potential selection bias.
    • Generalizability beyond the current setting.
Comments on the Quality of English Language

Present form of writing enough to publish 

Author Response

Thank you for your review and comments.

  1. Introduction: While the abstract and conclusion indicate a strong rationale, the introduction in the main text should ensure that: The gap in literature is clearly established. Differences between PBT and conventional radiotherapy are highlighted. The need for qualitative exploration is justified.

This has been actioned within the text.

  1. Methodology Suggestions: Clarify sample size justification. Include ethical approval. Define inclusion and exclusion criteria. Explain how themes were derived and validated.

This has been actioned within the text.

  1. Results clearly presented and supported by data? Five well-defined themes are presented. Findings are rich and nuanced, e.g., issues of emotional wellbeing, financial stress, and abrupt decline in healthcare contact post-treatment. Suggestion: Include patient quotes in the results section of the full text to enhance the authenticity of the themes.

Extensive quotes are already provided in both Table S1 and directly within the text after each theme.  

  1. Limitation: Stated that this is the first in-depth report on the topic, suggesting novelty. However, the limitations section could elaborate on: Potential selection bias. Generalizability beyond the current setting.

This has been actioned within the text.

Reviewer 2 Report

Comments and Suggestions for Authors

Hi

1- introduction: can you expand more why you focused on both radiotherapy and PBT. 

2- Introduction:  discuss the gaps in available qualitative studies that this study evaluates,

3-Introduction:  hidden burdens?.. please explain more

4-Methods:  can you discuss the data saturation based on selected samples. 

5-Methods:  how the coding is performed and validated. please discuss more inter-coder reliability or triangulation ? 

6- Methods:More discussion on Otter AI: validation, accuracy, limitation, reliability,,, 

7-discussion: how this study's findings can directly inform service redesign or policy.

Author Response

Thank you for your review and comments.

  1. introduction: can you expand more why you focused on both radiotherapy and PBT.

This has been addressed in the text: ‘The photon radiotherapy and PBT service at our large inner-city university hospital operates with a shared, rotational workforce and offers comparable levels of clinical support to patients across both treatment modalities. Importantly, there is often clinical overlap between the two pathways; for example, patients receiving PBT frequently have a backup photon radiotherapy plan in place in case of machine breakdowns or other technical issues. Given this interconnectedness, it was important to explore both treatment experiences to gain a comprehensive understanding of patient journeys within the service.’

  1. Introduction: discuss the gaps in available qualitative studies that this study evaluates,

    This has been addressed in the text: ‘Existing qualitative studies which do explore patient perspective often focus on in-formation needs, survivorship, or symptom assessment[3, 10], with limited attention to the holistic patient experience and perceived burden.[2, 3]’

  2. Introduction: hidden burdens?.. please explain more

This has been addressed in the text: ‘Logistical concerns such as managing time off work or school, financial pressures such as the direct costs of transport or reduced income, and interpersonal impacts such as shifts in family dynamics or roles within the family have all previously been reported, but rarely taken into account when exploring cancer treatment toxicity.’

  1. Methods: can you discuss the data saturation based on selected samples.

    This has been addressed in the text: ‘Data saturation was actively monitored throughout the analysis process and was considered achieved when no new themes or insights emerged from the final interviews, supporting the adequacy of the sample size.’

  2. Methods: how the coding is performed and validated. please discuss more inter-coder reliability or triangulation ?

    This has been addressed in the text: ‘Investigator triangulation was employed to increase credibility of research findings. Data saturation was actively monitored throughout the analysis process and was considered achieved when no new themes or insights emerged from the final interviews, supporting the adequacy of the sample size.’

  3. Methods:More discussion on Otter AI: validation, accuracy, limitation, reliability,,,

    This has been addressed within the text: ‘Otter AI was chosen due to its high reported transcription accuracy for English-language recordings in healthcare and research settings. However, recognising that automated transcription tools can be limited in distinguishing between speakers, handling overlapping dialogue, and capturing medical or technical terminology, each transcript was manually reviewed and edited to ensure verbatim accuracy.’

  4. discussion: how this study's findings can directly inform service redesign or policy.

This has been addressed in the conclusion: ‘Still, these insights will directly inform the design of the service by guiding the development of more patient-centred care pathways, such as improved communication strategies, enhanced psychosocial support, and more flexible scheduling to reduce travel and waiting times. At a policy level, the findings support the case for greater investment in equitable access to advanced radiotherapy technologies like PBT, and for policies that recognise the holistic burden of treatment, not just clinical outcomes.’

Reviewer 3 Report

Comments and Suggestions for Authors

Dear authors, congratulations on your work, but here are some suggestions for improving your manuscript:
Line 41: you need to explain better what you mean by ‘logistical, financial and interpersonal impacts’. What do you mean?
Line 49: ‘There has been significant research within radiation oncology over recent years that aims to reduce physical treatment-related toxicities’. Please cite these studies in the manuscript.
The introduction is missing a section describing radiotherapy and PBT. You need to describe the two techniques in detail and help readers understand the physical differences between them.
The discussion section should include references to articles that have dealt with a similar topic or that are similar to the holistic approach of this article.
The limitations of the studies should be placed in a separate section.
The bibliography should be expanded. 

Author Response

Thank you for your review and comments.

  1. Line 41: you need to explain better what you mean by ‘logistical, financial and interpersonal impacts’. What do you mean?
    Example added into introduction: Logistical concerns such as managing time off work or school, financial pressures such as the direct costs of transport or reduced income, and interpersonal impacts such as shifts in family dynamics or roles within the family have all previously been reported, but rarely taken into account when exploring cancer treatment toxicity.
  2. Line 49: ‘There has been significant research within radiation oncology over recent years that aims to reduce physical treatment-related toxicities’. Please cite these studies in the manuscript.
    This has been actioned.
  3. The introduction is missing a section describing radiotherapy and PBT. You need to describe the two techniques in detail and help readers understand the physical differences between them.
    This has been actioned.
  4. The discussion section should include references to articles that have dealt with a similar topic or that are similar to the holistic approach of this article.
    This has been actioned.
  5. The limitations of the studies should be placed in a separate section.
    This has been actioned.
  6. The bibliography should be expanded.
    This has been actioned.

Reviewer 4 Report

Comments and Suggestions for Authors
  1. Line 52 - To be statistical correct, whenever mentioned the word “significant” – please report the corresponding p-value to support such statement with the word of “significant”. If no p-value could be reported – suggest replacing with other words or phrases.

  1. Line 87 – What is the theory behind “aim was to recruit 30 patients”? What were the characteristic difference (like age, cancer stages,.. etc.) between patients taken (n=20) and rejected the interview (n=10)?

  1. Table 2. Participant characteristics – it would be very interesting shown 20 participants and 10 non-participants (added) – readers could compare how different between them.

  1. Was there power analysis conducted for the study? What was the statistical power for 20 patients (Five themes were identified) participating the study?

  1. One major fraud for this study is the super small sample size (N=20). The complexity of the research questions needs much larger sample size to provide statistical valid answers.  Authors did not mention study limitation of insufficient statistical power (due to super sample size) to draw meaningful conclusion.

Author Response

Thank you for your review and comments. Please see our responce below:

  1. 52 - To be statistical correct, whenever mentioned the word “significant” – please report the corresponding p-value to support such statement with the word of “significant”. If no p-value could be reported – suggest replacing with other words or phrases.

    Adjusted in text – but the word significant was used in the introduction only, which fits the narrative and referenced literature.

  2. Line 87 – What is the theory behind “aim was to recruit 30 patients”?

As outlined in previous literature, sample sizes in qualitative studies are typically small and purposively selected, with the goal of achieving data saturation—the point at which no new themes or insights emerge from the data. In this study, we conducted 20 in-depth interviews, which is well within the accepted range for qualitative health research.

Vasileiou, K., Barnett, J., Thorpe, S. et al. Characterising and justifying sample size sufficiency in interview-based studies: systematic analysis of qualitative health research over a 15-year period. BMC Med Res Methodol 18, 148 (2018). https://doi.org/10.1186/s12874-018-0594-7

  1. What were the characteristic difference (like age, cancer stages,.. etc.) between patients taken (n=20) and rejected the interview (n=10)? Table 2. Participant characteristics – it would be very interesting shown 20 participants and 10 non-participants (added) – readers could compare how different between them.

    This is not something that is usually reported in qualitative research. We have put some basic characteristics in the text but we will not include them in the table because this is a qualitative study and comparisons between participants and non-participants is not relevant.

  2. Was there power analysis conducted for the study? What was the statistical power for 20 patients (Five themes were identified) participating the study? One major fraud for this study is the super small sample size (N=20). The complexity of the research questions needs much larger sample size to provide statistical valid answers. Authors did not mention study limitation of insufficient statistical power (due to super sample size) to draw meaningful conclusion.

We appreciate the reviewer’s feedback, but respectfully clarify that this study employed a qualitative methodology, and therefore a statistical power analysis is not applicable. The aim of qualitative research is not to generalise findings to a larger population through statistical inference, but to explore depth, context, and meaning within a particular phenomenon or experience.

Reviewer 5 Report

Comments and Suggestions for Authors

The authors present a study of patients who have undergone proton beam radiotherapy with a very impressive portrayal of the psychological processing of cancer patients.

The text is well structured, the introduction and methods are clearly presented. Results and discussion sections are clear.

I only have minor comments:

As a suggestion, the authors could add to the information on the patient population whether the tumors were metastatic or localized, since a metastatic stage may have a stronger influence on well-being regardless of the type of therapy.

Please correct 3.1 „infromational…“

Add acknowledgements or remove template.

Author Response

Thank you for your review and comments.

  1. I only have minor comments: As a suggestion, the authors could add to the information on the patient population whether the tumors were metastatic or localized, since a metastatic stage may have a stronger influence on well-being regardless of the type of therapy.

    Thank you for this suggestion. Please see added sentence: ‘All patients were required to have localised disease in order to participant, as metastatic stage may have a stronger influence on wellbeing regardless of treatment type.’

  2. Please correct 3.1 „infromational…

    Thank you for highlighting this.

  3. Add acknowledgements or remove template.

    Thank you for highlighting this.

Round 2

Reviewer 3 Report

Comments and Suggestions for Authors

Dear authors, you have followed my advice and greatly improved your manuscript. Congratulations!